# Risk of Malignant Transformation in 3173 Subjects with Histopathologically Confirmed Oral Lichen Planus: A 33-Year Cohort Study in Northern Italy

**DOI:** 10.3390/cancers13225740

**Published:** 2021-11-16

**Authors:** Paolo G. Arduino, Alessandro Magliano, Alessio Gambino, Alessandra Macciotta, Mario Carbone, Davide Conrotto, Dora Karimi, Marco Carrozzo, Roberto Broccoletti

**Affiliations:** 1Department of Surgical Science, Oral Medicine Section, CIR-Dental School, University of Turin, 10126 Turin, Italy; alessandro.maglia@edu.unito.it (A.M.); alessio.gambino@unito.it (A.G.); mario.carbone@unito.it (M.C.); davide.conrotto@unito.it (D.C.); dora9k@hotmail.it (D.K.); roberto.broccoletti@unito.it (R.B.); 2Department of Clinical and Biological Sciences, University of Turin, 10126 Turin, Italy; alessandra.macciotta@unito.it; 3Department of Oral Medicine, School of Dental Sciences, University of Newcastle upon Tyne, Newcastle upon Tyne NE2 4BW, UK; marco.Carrozzo@newcastle.ac.uk

**Keywords:** lichen planus, malignant transformation, clinical features, treatment, outcome, HCV, survival rate

## Abstract

**Simple Summary:**

Up to 1–4% of OLP patients could develop oral cancer, and identifying factors that could help in early detection could be extremely useful. The aim of our cohort study was to assess the clinical characteristics of one of the biggest populations ever reported with a histologically confirmed diagnosis of OLP. We identified that OLP patients who developed oral cancer usually underwent less frequent periodical follow-up visits, and older age increased the risk of death. As a significant number of OLP patients have a risk of malignant transformation, there is a critical need to review these patients preferentially, by trained clinicians, at least once a year, possibly for a lifetime.

**Abstract:**

Background: Oral lichen planus (OLP) is considered an oral potentially malignant disorder. The aim of our study was to estimate the risk for oral cancer in patients diagnosed with OLP. Methods: A population-based cohort study between January 1988 and December 2020 at one hospital in Northern Italy was performed. The primary endpoint of the study was that of the histopathological diagnosis of oral cancer during the follow-up period. Results: The study population comprised 3173 patients. During the follow-up period, 32 men and 50 women developed an oral squamous cell carcinoma (2.58%), with a mean time of 103.61 months after the initial diagnosis of OLP, and 21 patients died because of oral cancer. Almost half of the deceased patients had the last follow-up visit before cancer diagnosis in a period of more than 12 months. Older age, having a red form of OLP and fewer sites of involvement, increased the risk of having cancer, while age and no treatment increased the risk of death. Conclusion: This is the largest group of OLP patients with such a long follow up ever reported. Due to the increased risk of having a malignant transformation, especially in elderly subjects, OLP patients should be regularly followed up, particularly in the Northern Italian population.

## 1. Introduction

Oral lichen planus (OLP) is a potentially malignant, chronic inflammatory disease, affecting nearly 2% of the population and causing a variety of oral lesions [1,2,3]. Its precise etiology remains unknown, but it possibly represents a cell-mediated immunological response to an induced antigenic change in predisposed patients [4]. OLP is likely not caused by a distinct antigen, assuming that previous works about OLP T-cell variable region genes have not discovered the presence of a limited number of them [5]. OLP is possibly influenced by a different range of extrinsic antigens, altered self-antigens, or other superantigens [6]. A causal association has also been reported between OLP and hepatitis C (HCV), with evidence that the virus could replicate in the oral mucosa and be involved in OLP pathogenesis [7]. In particular, during the 1990s, data reported from our group described a prevalence of HCV antibodies in OLP patients (27.1%) as significantly higher than in control cases (4.3%), but in the last decade, we reported a smaller overall prevalence (2.5%) of anti-HCV-positive patients [8]. This could be related to an unswerving decrease in the HCV prevalence reported in European countries in the last decade, indicating that the epidemiology of HCV has distinctly changed.

As said, the most severe complication of OLP is the development of an oral squamous cell carcinoma (OSCC), although this is still a very controversial matter. A recent systematic review reported a combined malignant transformation rate of 1.14% for OLP (95% CI = 0.84–1.49), also stating that these data could be underestimated due to essentially too restrictive diagnostic criteria, inadequate follow-up periods, and/or low quality of studies [9]. Differently, other authors have reported a malignant transformation rate of less than 0.5%, suggesting that the reported OLP malignant transformation rates are exaggerated, and they do not reflect the actual clinical course of the disease, particularly when strict clinical and histopathological criteria are applied [10].

At the end of the 1980s, our group started to collect data to estimate the potential risk of malignant transformation in a Northern Italian cohort of OLP patients, and we published the first results in late 2004 [2]. We were able to initially show that OLP patients had a significantly increased risk of oral squamous cell carcinoma (OSCC), and HCV infection apparently increased proportionally this risk. In this new study, we report our evaluation after 33 years of observation. To the best of our knowledge, this is the largest number of OLP patients ever reported with such a follow-up period. Based on these results, the current concepts for the malignant potential of Northern Italian OLP are critically analyzed.

## 2. Materials and Methods

### 2.1. Study Design

We conducted a population-based cohort study examining patients with OLP diagnosed between January 1988 and December 2020 at one hospital in Northern Italy. At baseline, we collected demographic information, including smoking (current or former smoker vs. nonsmoker), alcohol consumption (current or former drinker vs. nondrinker), clinical aspect of the lesions, sites of involvement, and HCV status. Subjects had to be residents in the Piedmont region, Northwest Italy. We decided to study the risk of oral cancer development, considering also commonly cited confounding factors. The work was approved by the ethical review board of the CIR-Dental School, University of Turin (CIR-PO-pga2020/09), which did not necessitate specific informed consent because of the nature of the study. The study was also registered on the ISRCTN registry (#ISRCTN47966874).

### 2.2. Data Collection

The individual data and outcome events were collected from a standardized computerized database [11] to a digital case report form. Three authors (P.G.A., D.K., and A.M. (Alessandro Magliano)) gathered all the patient data.

### 2.3. Participants

All adult patients examined during the inclusion period were enrolled. A final diagnosis of OLP was based on the following criteria:A.Clinical features: the presence of characteristic bilateral papular and/or reticular lesions (Wickham striae), alone or in association with atrophic or erosive lesions; lesions in close contact to oral amalgam restorations (suspected as “oral lichenoid contact lesions”) were excluded from the analysis; unilateral lesions in patients under new medication in the previous 6 months (which could be theoretically considered “causative” or so-called oral lichenoid drug reactions) were also excluded.B.Histopathological features: hyperkeratosis, either ortho- or parakeratosis; well-defined “band-like” zone of lymphocytic infiltration in the upper layer of the stroma: evidence of “liquefaction degeneration” of the lower layer of the epithelium, and absence of epithelial dysplasia [12].

### 2.4. Covariates

The clinical forms of OLP were divided into: (a) “white form”, which included the papular, reticular, and plaque forms; (b) “red form”, which included all the atrophic or erosive lesions, irrespective of a simultaneous presence of a white form. For symptomatic patients, corticosteroids therapy, in association with antimycotics, was given as previously published [11]. If subjects underwent at least one treatment, they were classified as “therapy positive”. Antibodies to HCV were measured for each patient after 1992 [11]. For patients who developed a malignant transformation in the follow-up period, the tumor grade was detailed according to the WHO classification (well, moderately, or poorly differentiated), and it was blindly and retrospectively re-examined by an expert oral pathologist. The cancer cases were also classified according to treatment modalities: surgery, radiotherapy, surgery in combination with radiotherapy, or no treatment. Surgery was the chief modality of treatment and patients with node-positive neck disease also underwent neck dissection in the same way as when tumor invasion of the midline structures was detected. Adjuvant radiotherapy with a local dose field of 55 to 72 Gy was used in patients with positive or close margins, vascular or perineural invasion, and extracapsular spread [11].

### 2.5. Outcomes

The primary endpoint of the study was that of the histopathological diagnosis of oral cancer. A latency time of at least 6 months between the diagnosis of OLP, and the diagnosis of the oral carcinoma was considered to exclude concurrent presentations; accordingly, the follow up began 6 months after the OLP diagnosis. Secondary outcome measures were considered as follows: tumor site of involvement, T classification, neck nodes association, treatment modalities, outcome, and survival rate.

### 2.6. Outcome Period

The start of the study period was defined as the day of the histopathological confirmation diagnosis of clinical OLP. The cohort was followed up to 31 December, 2020; if earlier, the end of the study period was defined as the first of the following events: death, diagnosis of an oral squamous cell carcinoma, or the impossibility to reattend the clinical visit due to personal reason. In the subgroup of oral cancer patients, we also collected data after the oncological diagnosis. Periodical visits for all patients were carried out with a frequency usually based on the clinical feature and the need for therapy; in general, patients were seen at least once a year. Whenever a malignant evolution was suspected, an incisional biopsy was performed.

### 2.7. Bias and Study Size

The risk of bias due to patient selection was restricted by the consecutive patient inclusion and lack of auto-selection. The aim of the study was to study all our patients to delineate the increase in risk for cancer development as precisely as possible; the sample size was not limited to the number of subjects needed to detect a substantial increase.

### 2.8. Statistical Analysis

Demographic and clinical characteristics were described by mean and standard deviation, median and interquartile range, or frequency and percentage, according to variable distributions. Additionally, appropriate statistical tests (*T*-test or Wilcoxon–Mann–Whitney or Chi-squared or Fisher’s exact tests) were applied to compare patients who developed oral cancer during the follow up with those who did not develop it and to compare cancers features between dead and survived cancer patients. Then, a multivariate Cox proportional-hazards (PH) model was performed for investigating the association between time-to-onset of cancer and the following factors of interest: gender, age at lichen diagnosis, type of lichen, number of sites involved, smoking habits, HCV, and corticosteroid therapy. In the subgroup of cancer patients, cumulative hazards and survival to 60 and 120 months after cancer diagnosis were computed. Furthermore, a multivariate Cox PH model was carried out for investigating the association between survival time and cancer-related variables (T, N, grading parameters, site, and treatment). All statistical analyses were carried out considering a first-type error α = 0.05 and using R software (version 4.0.5).

## 3. Results

Among 3872 subjects initially screened for a possible diagnosis of OLP, 699 were excluded because of the lack of histopathological confirmation or because of a follow up inferior to 12 months. Finally, a total of 3173 OLP were analyzed, of whom 2001 were women (f:m = 1.71:1). The mean age at presentation was 59 years (SD = 12.65). Follow up ranged from 12 to 384 months (median follow up of 48 months). The clinical features of total subjects are reported in Table 1, also showing the different characteristics of oncological cases and those who were free from cancer during the study.

During the follow-up period, 32 men and 50 women developed an oral squamous cell carcinoma (2.58%), with a median time of 96 months after the initial diagnosis of OLP (Figure 1 and Figure 2).

The mean age of these patients was 60.78 years (SD = 10.31). Patients who underwent a malignant transformation have been followed up for a longer period (*p* < 0.001). Most of the OLP patients were nonsmokers (86.1%), with an even bigger percentage in cancer patients (91.5%) but not statistically significant. Regarding HCV status, 193 patients were infected: this positive status was statistically more prominent in the cancer group (*p* = 0.031). According to the OLP clinical OLP form, 1893 patients (59.7%) had white lichen and 1280 the red form (40.3%); a higher percentage of red lichen has been described in cancer cases but without statistical significance. Although most patients had multiple oral sites of involvement, cancer patients developed OLP lesions in a statistically smaller number of sites (*p* = 0.018). The buccal mucosa was the most common location in each form (88.8%), followed by the tongue (59%) and gingiva (41.6%). The gingiva involvement was less reported in oncological patients (*p* = 0.012). Cancer patients underwent previously medical treatment similarly to others.

Table 2 reports the distinctive features of cancer patients, also comparing those who died in the follow-up period, specifically 21 subjects, (25.6%) and those who were still alive. Deceased patients were statistically older (*p* = 0.010), and they had a more frequent, but not statistically significant, form of red lichen. Moreover, in terms of gender, smoke exposure, and HCV status, the two groups were similar. Deceased patients also statistically showed bigger tumors and more diffuse node involvement (*p* = 0.013, and *p* = 0.019, respectively). In general, on the biopsy specimen, 40 patients were identified as well differentiated, 32 as moderately differentiated, and 6 as poorly differentiated. Four patients were identified as carcinoma in situ. Deceased patients had bigger oncological lesions (*p* = 0.013) and showed more neck involvement (*p* = 0.019). The tongue was the site most affected (39%), followed by the buccal mucosae (30.5%) and the gingiva (20.7%). Surgery was the primary modality of treatment, performed in 76 patients; postoperative radiation therapy was provided to 19 patients (after surgery) and to 3 patients as elective; only 3 patients did not undergo any therapy. The interval (in months) from the last follow-up visit before the histopathological cancer diagnosis was also recorded, and we reported a statistical difference between deceased and alive patients (*p* = 0.016), the last ones usually seen in the first 6 months after the last regular visit.

Almost half of the deceased patients had the last follow-up visit before cancer diagnosis in a period of more than 12 months. The median time for patients with a diagnosis of a tumor T1 or T2 and N1 was 6 months, while for patients with T3 or T4 and *n* > 1, it was 36 months, with a statistically significant difference (*p* = 0.018).

The multivariate Cox PH model, for investigating the association between time-to-onset of cancer and all the factors of interest (Table 3), revealed that older age, having a red form of OLP, and fewer sites of involvement, increased the risk of developing an OSCC. HCV status increased the risk without statistical difference. The smoking habits and a previous treatment seemed to decrease the risk but not statistically.

Differently, the multivariate Cox PH model for investigating the association between survival time and cancer-related variables among cancer (Table 4) showed that older age and having performed no treatment at all increased the risk of death (*p* = 0.02 and *p* = 0.04, respectively). More than 6 months since the last visit of follow up increased the risk of death, but not statistically (*p* = 0.73), such as a bigger size of lesions, neck involvement, and worse histopathological grading.

Finally, the reported survival rate was 82% at 60 months and 68% at 120 months (95% CI = 0.73–0.91 and 95% CI = 0.56–0.82, respectively) (Figure 3).

## 4. Discussion

OLP is currently considered an oral potentially malignant disorder (OPMD), although its malignant transformation rate is controversial, largely because of the lack of specific, worldwide agreed, diagnostic criteria [9]. In the last 3 decades, many authors have tried to analyze the factors related to the possible modalities of cancerization. However, even though OLP malignant transformation is probably the most important disease complication, to date, there is still little knowledge about the clinical behavior and prognosis of these cancers [13,14,15,16,17].

Very recently, a systematic review of systematic reviews recorded a 0.44% to 2.28% rate of malignant transformation of OLP [18], similar but slightly smaller than the value found in our investigation (2.58%). However, different from many other papers about this topic, the cases reported in our analysis fulfilled strict clinical and histological diagnostic criteria to avoid bias as much as possible. Moreover, the present study, reporting the largest cohort of patients with OLP followed up for a period of 33 years, found that the malignant transformation occurred in a mean time of 103.61 months after the initial diagnosis of OLP; such a long follow up has rarely been documented. It is also remarkable that in our cohort, just a minimal percentage of patients showed complete remission of clinical OLP signs [11]. For all these reasons, our findings reasonably seem to suggest that OLP patients should be followed up over an extensive period of time, possibly lifelong, although there is a lack of information to support any strong recommendations on the appropriate periodicity of follow-up sessions [9,10]. According to our data form, it might be appropriate to organize routine examinations at least on an annual basis, especially for elderly patients. In fact, according to our data, patients who are later to the follow-up controls can recur with more extensive and more aggressive cancer lesions. Such lesions therefore placed the patient at greater risk of not being able to undergo specific treatments, and this increases the risk of death in a statistically significant way (see Table 4), especially in elderly subjects.

As reported previously, most patients in our OLP cohort showed no increased prevalence of cigarette smoking [11]. However, this habit has been reported as one that could influence the risk of OLP malignant transformation. Our analysis does not support this view, and this could be explained by differences between our population and others from different parts of the world and because of the limited number of smoking patients (*n* = 7) who developed an oncological event. However, as tobacco and excessive alcohol consumption are the main risk factors for OSCC, it is always important, at least, to discuss the issue with OLP patients when it is relevant.

The percentage of OLP patients who were HCV positive and developed an OSCC was higher than in HCV negative ones, and this status increased the risk of cancer even if not statistically. However, the proportion of OLP with chronic HCV infection in our cohort has shown a steady decrease in the last 2 decades [2]. This fact, together with the availability of new very effective direct antiviral therapies [19], suggests that in the coming years, HCV infection could have a reduced clinical impact on OLP patients, at least in our geographic area.

Our study showed that white OLP could also progress to oral cancer, although it has recently been published [9] that pure reticular OLP does not meta-analytically progress to cancer. Our study cannot confirm this aspect as the malignancy rate of those cases presenting, as pure reticular OLP was not analyzed, and we recognize that plaque lesions may probably have a higher risk of malignancy. This aspect should be clarified in future studies.

Our evaluation did not find a statistical difference between OLP and OSCC cases for the site of lesions and for the atrophic-erosive form (red lichen), as compared to the white form, even if the multivariate analysis showed that the red form put the patients at greater risk of malignant transformation. Risk factors for malignant transformations were reported in the literature to be the location of OLP lesions (margins of the tongue) and clinical features of lesions (red color, heterogeneity) [14,18]. Moreover, for the first time in the literature, we also reported that fewer OLP-involved oral sites pose a higher risk of cancer development. This claim could be explained by the fact that less mucous membrane involvement may lead patients to less attention, but this should be further assessed with specific research.

The use of immunosuppressants has been regarded as a possible risk factor of malignant transformation, but it is unclear if this could be related to the suppression of the immune response or because of the decreasing local inflammatory reaction [20,21,22]. Moreover, topical corticosteroid, the gold-standard treatment for the last 20 years, could influence the time of cancer development [23,24]. Regarding the OLP treatment in our series, previously provided immunosuppressive therapy (topical and/or systemic) did not seem to influence the risk for malignant transformation. However, the number of treated patients, less than 30% in the present cohort, is lower than data reported worldwide and compared to our previous studies. This could be possible due to a change in OLP clinical presentation in the last years, but further prospective studies should confirm those data.

When performing research about OLP, as said, rigorously diagnostic criteria guided by both pathological and clinical characteristics should be sustained to differentiate studied disease from other similar disorders, hence warranting that selection biases should be prevented. This is particularly important when studying its malignant transformation pattern because other OPMDs, such as proliferative verrucous leukoplakia (PVL) above all, or oral lupus erythematosus and oral lichenoid lesions. In particular, PVL, which carries a higher malignant transformation rate, may share some clinical and histologic characteristics with OLP but express dissimilar clinical comportment [25,26,27,28]. Distinguishing OLP and PVL could be more challenging, as the latter is reported to show lichenoid lesions specifically during the initial stages; however, the presence of multifocal verrucous characters, as the development of multiple oral cancers over time, would ultimately point towards a diagnosis of PVL [29]. None of the cancer cases reported showed this peculiar feature. Moreover, the consideration of dysplasia as a diagnostic exclusion criterion is very controversial, and numerous authors are against it; our results indicate that even when excluding cases of dysplasia, which would be the most likely to develop cancer, the malignancy rate we obtained is very high. It is even possible that this rate is underestimated by the exclusion of dysplasia cases.

Finally, we decided to make a comparison of survival with data previously reported by our group, but in patients without signs of OLP [29]. Comparing survival rates with a similar population by age and ethnicity, it was possible to deduce that patients who developed cancer after diagnosis of OLP seemed to have a slightly higher chance of survival, with survival rates of 82% and 68% (at 60 months and 120, respectively) [29]. Our previous data about the survival rates in oral cancer in a naïve population described a survival rate of 77% at 60 months and 59% at 120 months. OLP-associated OSCC has already been reported as associated with a better survival rate, also very recently [30,31].

## 5. Conclusions

This is the largest group of OLP patients with such a long follow up ever reported. The risk of malignant transformation was found to be increased for elderly patients with a red form and fewer sites of involvement, regardless of gender, smoking status, HCV status, and previous therapies. The risk of death was higher in elderly patients with more aggressive forms who did not undergo any treatment. The reported survival rate, however, was slightly better if compared to those of a similar population without signs of OLP. As Northern Italian OLP patients have an increased risk of malignant transformation, there is a need to review all these patients by trained clinicians, at least once a year, possibly for a lifetime. It would also be advisable to instruct patients to anticipate the visit if they were to show changes in shape, volume, or associated pain that do not respond to normal topical therapy for more than 15 days.

## Figures and Tables

**Figure 1 cancers-13-05740-f001:**
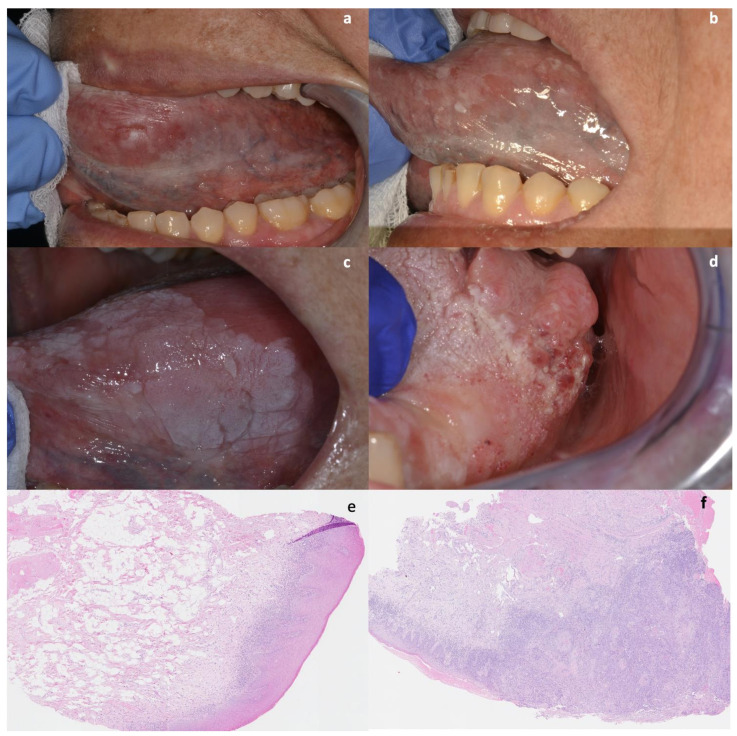
Evolution of OLP lesions on the left lateral border of the tongue of a female patient (**a** = 2012; **b** = 2014; **c** = 2015; **d** = 2017, date of cancer diagnosis; **e** = initial histopathological evaluation in 2021, **f** = OSCC diagnosis in 2017). This patient had also at onset typical reticular lesions on buccal mucosae.

**Figure 2 cancers-13-05740-f002:**
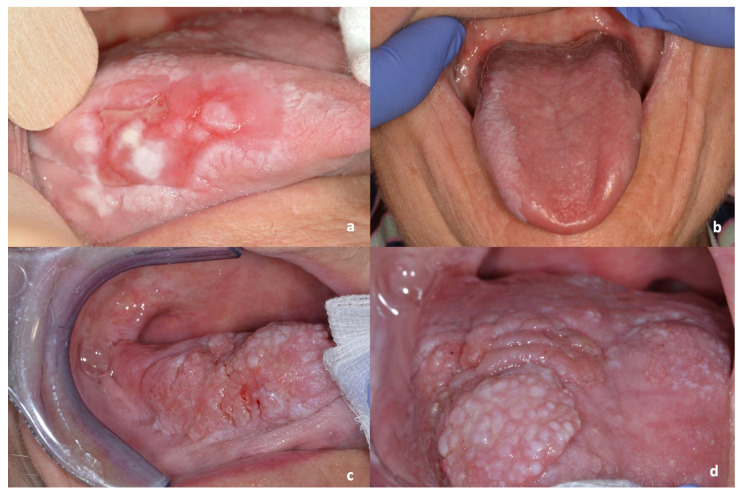
Evolution of OLP lesions on the right lateral border of the tongue and dorsum of the tongue of a female patient (**a** = 2009; **b** = 2010; **c** and **d** = 2015, date of cancer diagnosis).

**Figure 3 cancers-13-05740-f003:**
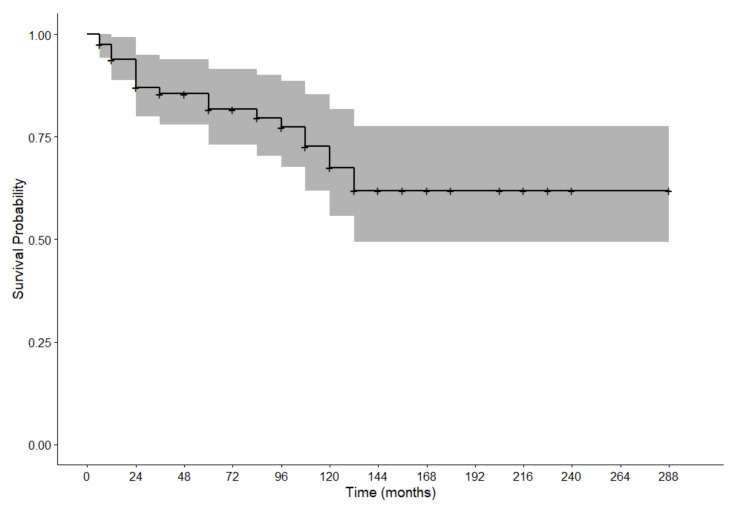
Survival curve after cancer occurrence.

**Table 1 cancers-13-05740-t001:** Demographic and clinical features of studied patients.

Variable		Overall	OLP Cases	Cancer Cases	*p*-Value
Number		3173	3091	82	
Gender					0.691 ^#^
	Male	1172 (36.9%)	1140 (36.9%)	32 (39.0%)	
	Female	2001 (63.1%)	1951 (63.1%)	50 (61.0%)	
Age (mean ± SD)		59.00 ± 12.65	58.95 ± 12.70	60.78 ± 10.31	0.118 ^$^
OLP clinical form					0.071 ^#^
	White	1893 (59.7%)	1852 (59.9%)	41 (50.0%)	
	Red	1280 (40.3%)	1239 (40.1%)	41 (50.0%)	
FU ^ (median [IQR])		48.00 [12.00, 120.00]	48.00 [12.00, 120.00]	96.00 [36.00, 144.00]	<0.001 *
Site of involvement					
	Cheek	2818 (88.8%)	2748 (88.9%)	70 (85.4%)	0.316 ^#^
	Tongue	1872 (59.0%)	1825 (59.0%)	47 (57.3%)	0.754 ^#^
	Gingiva	1319 (41.6%)	1296 (41.9%)	23 (28.0%)	0.012 ^#^
	Floor	106 (3.3%)	103 (3.3%)	3 (3.7%)	0.754 ^&^
	Palate	388 (12.2%)	381 (12.3%)	7 (8.5%)	0.301 ^#^
	Lips	273 (8.6%)	270 (8.7%)	3 (3.7%)	0.106 ^#^
NS ^ (median [IQR])		2.00 [1.00, 3.00]	2.00 [1.00, 3.00]	2.00 [1.00, 2.00]	0.018 *
Smoking status					0.155 ^#^
	Positive	441 (13.9%)	434 (14.0%)	7 (8.5%)	
	Negative	2732 (86.1%)	2657 (86.0%)	75 (91.5%)	
OLP therapy		838 (26.4%)	812 (26.3%)	26 (31.7%)	0.270 ^#^
HCV status					0.031 ^&^
	Positive	193 (6.1%)	183 (5.9%)	10 (12.2%)	
	Negative	2980 (93.9)	2908 (94.1%)	72 (87.8%)	

^ FU: months of follow up; NS: number of oral sites involved; ^#^ Chi-squared test; ^$^ *T*-test; * Wilcoxon–Mann–Whitney test; ^&^ Fisher’s exact test.

**Table 2 cancers-13-05740-t002:** Demographic and clinical features of patients who developed a malignant transformation.

Variable		Overall	Alive Patients	Deceased Patients	*p*-Value
Number		82	61	21	
Gender					0.097 ^#^
	Male	32 (39.0%)	27 (44.3%)	5 (23.8%)	
	Female	50 (61.0%)	34 (55.7%)	16 (76.2%)	
Age (mean ± SD)		60.78 ± 10.31	59.43 ± 11.06	64.71 ± 6.46	0.010 ^$^
OLP clinical form					0.077 ^#^
	White	41 (50.0%)	34 (55.7%)	7 (33.3%)	
	Red	41 (50.0%)	27 (44.3%)	14 (66.7%)	
Site of cancer					0.232 ^&^
	Cheek	25 (30.5%)	19 (31.1%)	6 (28.5%)	
	Tongue	32 (39.0%)	27 (44.3%)	5 (23.8%)	
	Gingiva	17 (20.7%)	11 (18.0%)	6 (28.6%)	
	Floor	1 (1.2%)	0	1 (4.8%)	
	Palate	2 (2.4%)	1 (1.6%)	1 (4.8%)	
	Lips	5 (6.2%)	3 (4.9%)	2 (9.5%)	
Smoking status					1 ^&^
	Positive	7 (8.5%)	5 (8.2%)	2 (9.5%)	
HCV status					1 ^&^
	Positive	10 (12.2%)	8 (13.1%)	2 (9.5%)	
T					0.013 ^&^
	1	55 (67.1%)	45 (73.8%)	10 (47.6%)	
	2	20 (24.4%)	13 (21.3%)	7 (33.3%)	
	3	2 (2.4%)	2 (3.3%)	0	
	4	5 (6.1%)	1 (1.6%)	4 (19.0%)	
N					0.019 ^&^
	0	65 (79.3%)	52 (85.2%)	13 (61.9%)	
	1	9 (11.0%)	6 (9.8%)	3 (14.3%)	
	2	7 (8.5%)	2 (3.3%)	5 (23.8)	
	3	1 (1.2%)	1 (1.6%)	0	
UCPD^@^ (median [IQR])		6.00 [4.00, 10.50]	6.00 [4.00, 6.00]	6.00 [6.00, 27.00]	0.037 *
UCPD^@^					0.016 ^&^
	1	58 (70.7%)	47 (77.0%)	11 (52.4%)	
	2	7 (8.5%)	6 (9.8%)	1 (4.8%)	
	3	4 (4.9%)	3 (4.9%)	1 (4.8%)	
	4	13 (15.9%)	5 (8.2%)	8 (38.1%)	

^@^UCPD = interval (in months) from the last follow-up visit before the histopathological cancer diagnosis was also recorded (1 = less than 6 months; 2 = from 7 to 12 months; 3 = from 13 to 23 month; 4 = more than 24 months); ^#^ Chi-squared test; ^$^ *T*-test; * Wilcoxon–Mann–Whitney test; ^&^ Fisher’s exact test.

**Table 3 cancers-13-05740-t003:** The Cox proportional-hazards model was performed for investigating the association between time-to-onset of a tumor and factors of interest (gender, age at lichen diagnosis, type of lichen, number of sites involved, smoking habits, hepatitis C virus, and corticosteroid therapy).

Variable		* HR	Lower 95%CI	Upper 95%CI	*p*-Value
Gender	Female vs. male	0.81	0.51	1.28	0.38
Age		1.04	1.02	1.06	0.00
Clinical type of OLP	Red vs. white	1.87	1.09	3.20	0.02
Number of involved sites		0.53	0.41	0.68	0.00
Smoking habits	Yes vs. no	0.75	0.34	1.66	0.49
Medical treatment for OLP	Yes vs. no	0.79	0.45	1.39	0.42
HCV status	Yes vs. no	1.67	0.85	3.29	0.14

* HR = hazard ratio; CI = confidence interval.

**Table 4 cancers-13-05740-t004:** The Cox proportional-hazards model conducted for investigating the association between survival time and cancer-related variables (T, N, grading parameters, site, and treatment).

Variable		HR *	Lower 95% CI	Upper 95% CI	*p*-Value
Gender	Female vs. male	2.55	0.69	9.43	0.16
Age		1.08	1.01	1.15	0.02
Months of the last visit before cancer diagnosis	>6 vs. ≤6	1.42	0.20	10.13	0.73
Type of treatment	Surgery vs. radiotherapy	0.89	0.16	4.86	0.90
Having done any type of treatment	No vs. yes	9.51	1.15	78.52	0.04
Neck surgery	1 vs. 0	0.83	0.21	3.22	0.78
T	3 + 4 vs. 1 + 2	5.48	0.56	53.51	0.14
N	2 + 3 vs. 0 + 1	1.82	0.33	10.14	0.49
G1	1 vs. 0	0.57	0.05	7.09	0.66
G2	2 vs. 0	1.34	0.12	14.83	0.81
G3	3 vs. 0	4.33	0.35	53.78	0.25

* HR = hazard ratio; CI = confidence interval.

## Data Availability

The data are freely accessible to qualified nonprofit investigators upon specific request.

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
