# Peer review of "Risk of Malignant Transformation in 3173 Subjects with Histopathologically Confirmed Oral Lichen Planus: A 33-Year Cohort Study in Northern Italy"

_cancers, 2021, doi:10.3390/cancers13225740_

Round 1

Reviewer 1 Report

Dear editor,

Thank you for the opportunity to review manuscript entitled „Risk of malignant transformation in 3,173 subjects with histo-2 pathologically confirmed oral lichen planus: a 33-year cohort 3 study in Northern Italy“. The study describes natural course of the large cohort of patients with OLP. It largely confirms the previous findings of similar studies. Although the number of patients and observation period are impressive, there are some issues with the manuscript to point out.

Major comments:

  • It is stated in the methods that patients that lesions were not related to oral amalgam restorations or drug intake. It is unclear whether these patients having used any drugs or having any amalgam restorations were excluded from the study or how these patients were handled
  • HCV status was evaluated after 1992 but all patients are included in Table 1 (193+2980), it is therefore unclear how many were in fact negative and how many not evaluated (before 1992) – this might in fact produce a huge bias in the results
  • Were all histopathologic findings retrospectively reviewed or did the authors rely on 30+ years old descriptions?
  • Spacing and rows in the tables should be better presented, it sometimes is a bit unclear which P value belongs to which variable
  • Discussion is rather short and only HCV is discussed in greater detail, in many parts too general (One of the most widely felt problems in oral medicine for patients diagnosed with OLP is how regularly periodic follow-up visits should be organized – as an example)
  • Conclusion statement: there is a need to review all these patients by trained clinicians, at least once a year – such recommendation cannot be made on reported results
  • There are only 16 references, out of which 10 references are more than 10 years old leaving only 6 up to date references which I find insufficient for such a large scale study (especially the discussion part is scanty)

Minor comments:

  • Many typo errors need corrections throughout the text – spelling and grammar check needed

Author Response

Thanks for your positive comments.

  1. It is stated in the methods that patients that lesions were not related to oral amalgam restorations or drug intake. It is unclear whether these patients having used any drugs or having any amalgam restorations were excluded from the study or how these patients were handled.
  2. We have treid to be more specific and changes the phrase as:

…..“Clinical confirmation with the presence of characteristic bilateral clinical signs [papular and/or reticular lesions (Wickham striae) alone or in association with atrophic or erosive lesions]; lesions in close contact to oral amalgam restorations (suspected as “oral lichenoid contact lesions”) were excluded from the analysis; singular lesions in patients under new medication in the previous 6 months (which could be theoretically considered as “causative”) were also excluded”…

  1. HCV status was evaluated after 1992 but all patients are included in Table 1 (193+2980), it is therefore unclear how many were in fact negative and how many not evaluated (before 1992) – this might in fact produce a huge bias in the results.
  2. Specific tests for hepatitis C have been available in Italy since the beginning of the 90’s, so our data on the issue are available since 1992. The few years not included in the analysis are however not influential.

  1. Were all histopathologic findings retrospectively reviewed or did the authors rely on 30+ years old descriptions?
  2. Yes, as clearly reported. This huge work started years ago, and the first references was this Gandolfo, S.; Richiardi, L.; Carrozzo, M.; Broccoletti, R.; Carbone, M.; Pagano, M.; Vestita, C.; Rosso, S.; Merletti, F. Risk of oral squamous cell carcinoma in 402 patients with oral lichen planus: a follow-up study in an Italian population. Oral. Oncol. 2004, 40, 77-83.

  1. Spacing and rows in the tables should be better presented, it sometimes is a bit unclear which P value belongs to which variable.
  2. We have tried to resolve this issue in a graphical manner but probably during the editor process It will be fixed.

  1. Discussion is rather short and only HCV is discussed in greater detail, in many parts too general (One of the most widely felt problems in oral medicine for patients diagnosed with OLP is how regularly periodic follow-up visits should be organized – as an example).
  2. We have tried to complete the discussion section with other topics as suggested.

  1. Conclusion statement: there is a need to review all these patients by trained clinicians, at least once a year – such recommendation cannot be made on reported results.
  2. We have changed part of the conclusion in order to be less specific and categorical.

  1. There are only 16 references, out of which 10 references are more than 10 years old leaving only 6 up to date references which I find insufficient for such a large-scale study (especially the discussion part is scanty).
  2. We have added some references as suggest by different reviewers.

Many typo errors need corrections throughout the text – spelling and grammar check needed.

  1. We have made all the corrections required and grammar was checked. Prof. Marco Carrozzo (a native Italian speaker but resident in UK since 2008 and Professor in oral medicine) has carefully reviewed all the paper.

Reviewer 2 Report

Dear Authors, thank you for your paper. It is very well written and very intersting although data are in accordance with a wide literature about LPO transformation...

Some questions should be advised: 

  • HCV infection was endemic in the mediterranean area but this was reduced in the last 10 years-- it should be addressed in the text and possibly correlated to a differences among patients
  • a paper on such topic should mandatorly contain a discussion about other diseases with similar clinical aspects, epecially PVL; also, the criteria used to exclude such diagnosis should be listed and discussed (please refer to Van der Wall,  Villa et al, Favia et al, etc etc), especially for the case as you showed in the figures, with the clinical features of PVL. This should be also supported by histological pictures as comaparison between the biopsies at different times.
  • references should be improved and not related only to authors studies 

Thank you for your paper

Author Response

Thanks for your positive comments.

  1. HCV infection was endemic in the Mediterranean area but this was reduced in the last 10 years-- it should be addressed in the text and possibly correlated to a differences among patients.
  2. This is right, and we have added some sentences about it. Moreover, we have discussed this also with our reference (Arduino, P.G.; Carbone, M.; Conrotto, D.; Gambino, A.; Cabras, M.; Karimi, D.; Broccoletti, R. Changing epidemiology of HCV infection in patients with oral lichen planus in north-west Italy. Oral. Dis. 2019, 25, 1414-1415).

  1. A paper on such topic should mandatory contain a discussion about other diseases with similar clinical aspects, especially PVL; also, the criteria used to exclude such diagnosis should be listed and discussed (please refer to Van der Wall, Villa et al, Favia et al, etc etc), especially for the case as you showed in the figures, with the clinical features of PVL. This should be also supported by histological pictures as comparison between the biopsies at different times.
  2. As suggested we have added some papers about PVL and discuss them. Histological pictures were added as required.

  1. references should be improved and not related only to authors studies 
  2. We have added some more recent references.

Reviewer 3 Report

The authors have shown their experience with handling data of this nature and how to report them appropriately making sure that all queries that could be raised from such data are already evaluated. This reviewer main concern is that the result of the Cox proportional hazard model presented for time to onset cancer and factors of interest as well survival time and cancer related variables are univariate. They have not explained if a multivariate analysis was done or how it was done. Without a multivariate analysis, it is difficult to accept that all the various factors studied actually have independent predictive (or prognostic) values or that they have the kind of strong influence the authors have attributed to them (regarding onset time or survival) throughout the manuscript. Another minor issue is that of occasionally using wrong choice of words, wrong spelling or misspellings etc. I flagged almost 20 in my first reading and would advise the authors to get the manuscript reviewed by a native English speaker who is well-versed in doing so for medical literature. In all the efforts of the authors are commendable and the quality of the work is good.

Author Response

  1. The authors have shown their experience with handling data of this nature and how to report them appropriately making sure that all queries that could be raised from such data are already evaluated. This reviewer main concern is that the result of the Cox proportional hazard model presented for time to onset cancer and factors of interest as well survival time and cancer related variables are univariate. They have not explained if a multivariate analysis was done or how it was done. Without a multivariate analysis, it is difficult to accept that all the various factors studied actually have independent predictive (or prognostic) values or that they have the kind of strong influence the authors have attributed to them (regarding onset time or survival) throughout the manuscript.

A.

We thank the reviewer for the comment. It gave us the opportunity to revise the text accordingly, specifying that we performed two multivariate Cox PH models. Indeed, after an initial descriptive analysis of individual factors, we built Cox models comprehensive of the variables considered.

We enlarged the comments related to the models, as well.

  1. Another minor issue is that of occasionally using wrong choice of words, wrong spelling or misspellings etc. I flagged almost 20 in my first reading and would advise the authors to get the manuscript reviewed by a native English speaker who is well-versed in doing so for medical literature. In all the efforts of the authors are commendable and the quality of the work is good.
  2. Prof. Marco Carrozzo (a native Italian speaker but resident in UK since 2008 and Professor in oral medicine) has carefully reviewed all the paper.

Round 2

Reviewer 1 Report

Thank you for revising this interesting manuscript. The text has improved a lot and overall quality of paper as well.

Minor comment: I cannot find citations 20,21,22,25 and 31 cited in the text - these should be added to correct places.

Author Response

We adjusted the references also in the text.